# Single-Stage Visual Relationship Learning using Conditional Queries

**Alakh Desai[1], Tz-Ying Wu[1], Subarna Tripathi[2], Nuno Vasconcelos[1]**
[1]University of California San Diego, USA
[2]Intel Labs, USA

## Abstract

Research in scene graph generation (SGG) usually considers two-stage models, that is, detecting a set of entities, followed by combining them and labelling all possible relationships. While showing promising results, the pipeline structure induces large parameter and computation overhead, and typically hinders end-to-end optimizations. To address this, recent research attempts to train single-stage models that are computationally efficient. With the advent of DETR[3], a set based detection model, one-stage models attempt to predict a set of subject-predicate-object triplets directly in a single shot. However, SGG is inherently a multi-task learning problem that requires modeling entity and predicate distributions simultaneously. In this paper, we propose Transformers with conditional queries for SGG, namely, **TraCQ** with a new formulation for SGG that avoids the multi-task learning problem and the combinatorial entity pair distribution. We employ a DETR-based encoder-decoder design and leverage conditional queries to significantly reduce the entity label space as well, which leads to $20\%$ less parameters compared to state-of-the-art single-stage models. Experimental results show that TraCQ not only outperforms existing single-stage scene graph generation methods, it also beats many state-of-the-art two-stage methods on Visual Genome dataset, yet is capable of end-to-end training and faster inference.

## 1 Introduction

Scene graph generation (SGG) provides a structured representation of visual relations in a scene. A scene graph is a set of *subject-predicate-object* triplets, where subjects and objects are entity nodes in the graph and predicates are edges representing the relationship between pairs of entities. Due to its compact structure, SGG has been adopted as a foundational step for several high-level machine cognition tasks, including caption generation [48, 47, 29], visual question answering [13, 30], image retrieval [14, 38] and image generation [15, 22, 36]. However, SGG is far from a trivial problem, due to the complexity of detecting and pairing subject-object entities and inferring the predicate between each subject-object pair, requiring the model to excel at both entity and relationship detection. Hence, most research in the problem decomposes it into two separate sub-tasks, entity detection and predicate detection, which are modeled sequentially. This leverages success in object detection [7, 31] and leads to the predominance of two-stage networks in the literature [4, 52, 35, 34, 21, 43, 45, 25], where an object detector such as the faster-RCNN [31] solves the first sub-task, reducing the SGG modeling to the second stage that addresses the pairing of subject-object entities and classifying of predicate categories. The predicate detection part is typically achieved by ranking the $\mathcal{O}(N^2)$ triplet proposals generated by the exhaustive pairing of $N$ entity predictions. While these models have shown promising results, the pipeline structure usually comes with significant parameter and computation overhead, which hinders end-to-end SGG optimization under memory constraints, and leads to sub-optimal solutions.

36th Conference on Neural Information Processing Systems (NeurIPS 2022).

To enable end-to-end optimization, some recent works introduce one-stage models based on point-based detection [26] and transformer-based detectors [5, 33, 20]. Since SGG is inherently a multi-task learning problem, the main idea is to model the entity detection and predicate detection in parallel, which can be done in multiple ways. These models, however, face the problem of having to learn a joint feature space for two very different tasks. To circumvent the difficulty of this problem, they disentangle entity detection and predicate classification by training separate decoder branches for the two tasks. However, we hypothesize that this form of disentanglement is too strong for SGG, since capturing the interactions in a scene requires three kinds of features: $f_e$ tuned for entity detection (object detection), $f_p$ tuned for predicate classification, and $f_i$ tuned for capturing the relationships between entities and their surroundings. While $f_e$ can be learned well under a complete disentanglement of the two tasks, $f_i$ is hard to capture without some amount of feature tuning for entity detection. In general, the task of predicate classification is what drives the learning of the interaction features. Therefore, $f_p$ and $f_i$ rely heavily on each other and benefit from some amount of coupled learning.

With this in mind, we propose Transformers with conditional queries (TraCQ), with a new formulation for scene graph generation. Instead of trying to learn $f_e$ and $f_i$ together in the entity detection branch, TraCQ learns $f_p$ and $f_i$ together in a predicate detection branch $\mathcal{H}$, which then conditions the learning of $f_e$ by a separate entity refinement module $\mathcal{C}$. A weak coupling between predicate and entity detection is then ensured by forcing $\mathcal{H}$ to learn $f_i$ through the prediction of loose estimates of subject and object bounding boxes. A major effect of this formulation is that the distribution that $\mathcal{H}$ learns is conditioned on the non-combinatorial predicate space, while typical SGG models disentangle the learning of $f_p$ from $(f_i, f_e)$ where the entity decoder is expected to learn a distribution based on the combinatorial $\mathcal{O}(\mathcal{E} \times \mathcal{E})$ space. Given that we have limited features and $(f_i, f_p)$ are closely related, it is easier to learn from the non-combinatorial predicate space than the $\mathcal{O}(\mathcal{E} \times \mathcal{E})$ entity pair space. This reduction in distribution space implies that a simpler and smaller model can be used for SGG.

Overall, this paper makes the following contributions. First, we introduce a new formulation of the single-stage SGG task that avoids the multi-task learning problem and the combinatorial entity pair space. Second, we propose a novel architecture, Transformers with conditional queries (TraCQ), wherein conditional queries are leveraged to significantly reduce the inference time and the distribution space to learn, which leads to 20% less parameters compared to state-of-the-art single-stage models. Finally, we show that TraCQ achieves improved SGG performance than state-of-the-art methods on Visual Genome dataset, yet is capable of end-to-end training and faster inference.

## 2 Preliminaries

**Transformers** [37] are powerful models for sequence modeling, based on an encoder-decoder architecture. Both the encoder and the decoder modules stack multiple attention-based blocks, differing on the type of attention mechanisms employed. Encoding blocks perform self-attention across input tokens, while decoding blocks perform cross-attention between encoder output and predictions. The attention operation is defined over a set of queries $\mathbf{Q} \in \mathbb{R}^{n \times d_q}$, keys $\mathbf{K} \in \mathbb{R}^{n \times d_k}$ and values $\mathbf{V} \in \mathbb{R}^{n \times d_v}$, where $d_q = d_k$, according to

$$\text{Attention}(\mathbf{Q}, \mathbf{K}, \mathbf{V}) = \text{Softmax}\left(\frac{\mathbf{Q}\mathbf{K}^T}{\sqrt{d_k}}\right)\mathbf{V}. \tag{1}$$

Information from different representation subspaces is learned at different positions with resort to multihead attention based on different attention heads,

$$\text{MultiHead}(\mathbf{Q}, \mathbf{K}, \mathbf{V}) = \text{Concat}(\text{head}_1, \cdots, \text{head}_h)\mathbf{W}^O \quad \text{where } \mathbf{W}^O \in \mathbb{R}^{d_v \times h} \tag{2}$$

$$\text{head}_i = \text{Attention}(\mathbf{Q}\mathbf{W}_i^Q, \mathbf{K}\mathbf{W}_i^K, \mathbf{V}\mathbf{W}_i^V), \tag{3}$$

where $\{\mathbf{W}_i^Q, \mathbf{W}_i^K, \mathbf{W}_i^V\}$ are the parameters of the $i^{th}$ head. This is followed by a normalization layer [2] (LN) with residual connection in each block. We denote the entire module as the attention stack, $Attn_{stack}(\mathbf{Q}, \mathbf{K}, \mathbf{V})$, of queries $\mathbf{Q}$, keys $\mathbf{K}$ and values $\mathbf{V}$, moving forward.

**DETR** [3] is a Transformer-based object detector. A CNN backbone first generates a feature tensor $\mathbf{F}_b \in \mathbb{R}^{H \times W \times d}$ for an image $\mathcal{I}$. An encoder then learns context features $\mathbf{F}_v = Attn_{stack}(\mathbf{Q}_E, \mathbf{K}_E, \mathbf{V}_E) \in \mathbb{R}^{L \times d}$, where $L = H \times W$, $\mathbf{Q}_E = \mathbf{K}_E = \mathbf{V}_E = \textit{flatten}(\mathbf{F}_b) + E_p$,

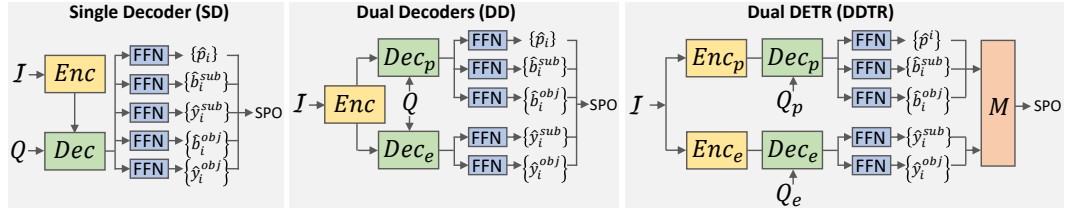

Figure 1: **Baselines**. $Enc$, $Dec$, and $M$ denote the encoder, decoder, and Hungarian matching respectively. $(\cdot)_p$ represents that the instance is for predicate detection, and $(\cdot)_e$ is for entity detection. SPO denotes triplets of <subject,predicate,object>.

and $E_p \in \mathbb{R}^{L \times d}$ is a fixed positional encoding. The decoder transforms a set of entity queries $\mathbf{Q}_{entity} \in \mathbb{R}^{N_e \times d_e}$ into entity representations

$$\mathbf{Z}_E = Attn_{stack}(\mathbf{Q}_{entity}, \mathbf{Q}_{entity}, \mathbf{F}_v) \in \mathcal{R}^{N_e \times d}. \tag{4}$$

Separate feed forward networks (FFNs) are finally used to predict each entity $\hat{e} = (\hat{b}, \hat{c})$, composed by class label $\hat{c}$ and a bounding box $\hat{b}$, from $\mathbf{Z}_E$.

DETR adopts a set prediction loss for entity detection, which performs bipartite matching between the ground truth set $Y$ and the predicted set $\hat{Y}$ with the Hungarian matching algorithm [19], i.e.

$$\hat{\sigma} = \arg\min_{\sigma \in \mathcal{G}_{N_e}} \sum_{i=1}^{N_e} \mathcal{C}_{match}(\hat{e}_{\sigma(i)}, e_i), \tag{5}$$

where $\mathcal{G}_{N_e}$ denotes the set of permutations of $N_e$ predicted entities, $e_i = (c_i, b_i) \in Y$, $c, b$ indicate target class and box coordinates respectively, $\hat{e}_{\sigma(i)} \in \hat{Y}$,

$$\mathcal{C}_{match}(\hat{e}, e) = -\mathbb{1}_{\{c \neq \phi\}}\mathbf{p}_{\hat{e}}^c + \mathbb{1}_{\{c \neq \phi\}}\mathcal{L}_{box}(\hat{b}, b) \tag{6}$$

and $\mathbf{p}_{\hat{e}}^c$ is the logit for class $c$ of entity $\hat{e}$. Note that $Y$ is padded with background tokens $\phi$. Finally, the set prediction loss is formulated as,

$$\mathcal{L}_{DETR} = \sum_{i=1}^{N_e}[\mathcal{L}_{cls}(\hat{c}_{\hat{\sigma}(i)}, c_i) + \mathbb{1}_{\{c_i \neq \phi\}}\mathcal{L}_{box}(\hat{b}_{\hat{\sigma}(i)}, b_i)], \tag{7}$$

where $\mathcal{L}_{cls}$ denotes the cross-entropy loss for label classification, $\mathcal{L}_{box}$ consists of $L_1$ and generalized IoU loss [32] for box coordinates regression.

In this work, we adopt a modified attention mechanism, namely Poll and Pool attention from PnP-DETR[39], which abstracts the image feature map into fine foreground object feature vectors and a small number of coarse background contextual feature vectors. We use PnP-DETR for faster convergence and computation efficiency.

## 3 Toward Single-Stage End-to-End Scene Graph Generation

### 3.1 Two-stage models

Given an image $\mathcal{I}$, scene graph generation produces a scene graph $\mathcal{G}$ to describe visual relations in a scene. $\mathcal{G}$ consists of a set of entity vertices $\mathcal{E}$, typically objects, and a set of directed edges $\mathcal{P}$ representing predicates, typically relationships between objects such as "to the left of." Each vertex in $\mathcal{E}$ is a tuple consisting of the bounding box $b$ and the class label $y$ of an entity instance, while each edge in $\mathcal{P}$ denotes the predicate label between a pair of entities. Each edge and its connected vertices form a relation tuple $< (b^{sub}, y^{sub}) - p - (b^{obj}, y^{obj}) >$, which is alternatively called *subject-predicate-object* (SPO) triplet, e.g. "a man in region $b^{sub}$ - to the left of - a car in region $b^{obj}$." This is a complex task involving entity and predicate detection, which requires to model both entity and predicate distributions. Prior research [52, 6] usually models the problem as

$$Pr(\mathcal{G}|\mathcal{I}) = Pr(\mathcal{Y}, \mathcal{B}|\mathcal{I})Pr(\mathcal{P}|\mathcal{B}, \mathcal{Y}, \mathcal{I}), \tag{8}$$

Table 1: **Preliminary experiments.** The architectures of these baselines are presented in Figure 1.

| Model | mean Recall (↑) | | | #params (↓) | Inference time (↓) |
| | @20 | @50 | @100 | (M) | (sec) |
|---|---|---|---|---|---|
| SD | 5.7 | 6.2 | 6.3 | 41.7M | 0.063 |
| DD | 6.5 | 6.9 | 7.0 | 51.1M | 0.068 |
| DDTR | 9.2 | 11.8 | 13.0 | 82.9M | 0.220 |

where $\mathcal{Y} = \{y_i\}_{i=1}^{|\mathcal{E}|}$ and $\mathcal{B} = \{b_i\}_{i=1}^{|\mathcal{E}|}$. A state-of-the-art object detector [7, 31] is then used to handle the first component, reducing the SGG problem to the second. While this two-stage formulation has produced good results, it leads to quite slow inference, since it requires an exhaustive search through the space of $\mathcal{O}(N^2)$. In addition, two-stage models usually have large parameter and computation overhead, which inhibits end-to-end SGG optimization, and makes the overall solution sub-optimal.

## 3.2    Learning scene graph generation with set predictions

With the advent of DETR [3], there has been a shift to single-stage end-to-end SGG learning, by adoption of a Transformer-based encoder-decoder architecture for SPO triplet set predictions. Several models have been proposed [5, 33, 20]. While efficient, these methods tend to have a weaker performance than two-stage models. We hypothesize that this is due to the entanglement between the feature spaces used to represent entities and predicates. This entanglement is unavoidable, since the detection of a predicate always requires some knowledge of the associated subject and object. However, a very strong entanglement is undesirable, because predicates and entities have very different distributions. Since the individual distributions of predicates and entities are long-tailed, the joint distribution of their pairs is extremely long-tailed. Hence, most pairs are very poorly represented in a highly entangled feature space.

We next test this hypothesis by performing some preliminary experiments on a set of baseline models of varying levels of disentanglement between the entity and predicate feature spaces.

**Single decoder (SD)** uses a pair of encoder and decoder modules, followed by five FFNs to decode each element $< (b^{sub}, y^{sub}) - p - (b^{obj}, y^{obj}) >$ of the SPO tuple. The encoder learns a context feature tensor from the input image $\mathcal{I}$, and the decoder takes these features and random queries $\mathcal{Q}$ to generate the FFN input. Since it generates the whole SPO triplet at once, this model learns a feature space where entities and predicates are highly entangled.

**Dual decoders (DD)** consists of a shared encoder and dual decoders for predicate and entity detection respectively, where the former decodes $< b^{sub} - p - b^{obj} >$ tuples and the latter decodes $< y^{sub} - y^{obj} >$ pairs. By feeding a common set of random queries $\mathcal{Q}$ to the two decoders, we ensure that no matching between the two output sets is needed. This is important to keep complexity competitive with that of SD. While introducing some additional cost, the use of two separate decoders encourages the disentanglement of the predicate and entity feature spaces. However, two architecture components still encourage entanglement: 1) the use of a common encoder and 2) the sharing of queries by the two decoders.

**Dual DETRs (DDTR)** avoids these problems by introducing separate DETR models, each with its own encoder, decoder and random queries, for detecting predicates and entities respectively. Similar to DD, one model decodes $< b^{sub} - p - b^{obj} >$, and the other $< y^{sub} - y^{obj} >$. Since each DETR has an individual set of random queries, a brute-forced matching between the two sets of outputs is needed to generate the detected graph. Assuming $M$ predicates and $N$ entity predictions, there are $M \times \mathbf{P}_2^N$ SPO tuples. Generating the cost matrix between the two sets is computationally expensive. However, this model also has the weakest entanglement between feature spaces.

In table 1, we compare the performance of these three baselines, in terms of mean Recall (see Section 5.1) on the Visual Genome (VG) [18] dataset. The number of parameters and the inference time are also provided. It is clear that disentangling predicate and entity representations is a good strategy for solving SGG. However, the computational cost of disjoint queries is quite high, including more parameters and lower inference speed. The question is whether the DD architecture can be improved to achieve performance closer to DDTR, while maintaining low parameter and computation overhead. To accomplish this, we next propose a model, TraCQ, based on the DD architecture. TraCQ

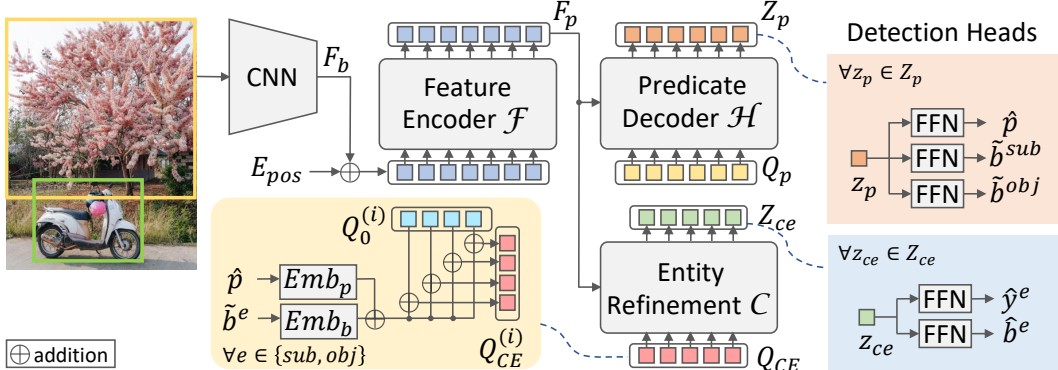

Figure 2: Model architecture of TraCQ.

reduces entanglement by replacing the shared queries of DD with conditional queries and leveraging decoupled training for predicates and entities.

## 4 Transformers with Conditional Queries

In this section, we introduce the proposed architecture, Transformers with conditional queries (TraCQ).

**Architecture** As shown in Figure 2, the TraCQ model employs the encoder-decoder Transformer design, composed of a feature extractor $\mathcal{F}$, a predicate decoder $\mathcal{H}$, and a conditional entity refinement decoder $\mathcal{C}$. The predicate decoder $\mathcal{H}$ predicts a set $\mathcal{S}_{pred}$ of $< b^{sub} - p - b^{obj} >$ tuples. The refinement decoder $\mathcal{C}$ predicts entity class labels and refines the bounding boxes detected by $\mathcal{H}$, to predict $< (b^{sub}, y^{sub}) - (b^{obj}, y^{obj}) >$ pairs. The idea is that, the use of $\mathcal{C}$ to refine the entity bounding boxes, frees $\mathcal{H}$ from solving this task with high accuracy. Hence, the feature space of $\mathcal{H}$ has to be less representative of entities, enhancing its disentanglement from the space of predicates. The details of each component are described below.

**Feature Extractor** The feature encoder $\mathcal{F}$ is a transformer encoder that takes a feature tensor $\mathbf{F}_b$ extracted from the image by the CNN backbone and a fixed positional encoding $E_{Pos}$ and implements the self-attention stack to produce a feature tensor $\mathbf{F}_p$ that encodes the global image context. To ensure that $\mathbf{F}_p$ emphasizes attention on scene relations, $\mathcal{F}$ is trained jointly with $\mathcal{H}$ and then frozen for the training of $\mathcal{C}$. This results in features that are optimal for predicate detection. These features are richer than those for entity detection, since predicate detection involves an understanding of the constituent entities in order to detect the relationship between them and even captures the interplay between the entities in the scene which is difficult to learn if $\mathcal{F}$ was trained with $\mathcal{C}$.

The feature $\mathbf{F}_p$ is used as the $\mathbf{K}$ and $\mathbf{V}$ values for both decoders $\mathcal{H}$ and $\mathcal{C}$. This module, therefore, acts as the common encoder for both decoders.

**Predicate Detection** The predicate detection module $\mathcal{H}$ lies at the heart of TraCQ. This module comprises of a decoder which complements the encoder from $\mathcal{F}$. Similar to an entity detector, it transforms a set of predicate queries $\mathbf{Q}_p \in \mathcal{R}^{N_p \times d_p}$ into the predicate representations $\mathbf{Z}_P = Attn_{stack}(\mathbf{Q}_p, \mathbf{Q}_p, \mathbf{F}_p) \in \mathcal{R}^{N_p \times d}$. Hence, the composition of $\mathcal{F}$ and $\mathcal{H}$ is equivalent to that of DETR. However, $\mathbf{Z}_P$ is fed to three distinct multi-layer FFNs that learn to predict the bounding boxes $\tilde{b}^{sub}$ and $\tilde{b}^{obj}$ and the predicate class $\hat{p}$.

In this way, $\mathcal{H}$ learns to predict not only the predicate label but also a pair of bounding boxes for the subject and the object. This requires $\mathcal{H}$ to both learn to localize related entities and understand the relationship between them. However, predicting two bounding boxes along with the predicate label is far from trivial, due to the entanglement problems discussed above. To reduce entanglement, $\mathcal{H}$ learns to only roughly localize the subject and object, leaving the exact bounding box localization to $\mathcal{C}$. Since the key goal of the DETR-like combination of $\mathcal{F}$ and $\mathcal{H}$ is to predict predicate categories

accurately, it learns a visually rich feature space for the predicate distribution. Overall, $\mathcal{H}$ predicts a set $\mathcal{S}_{pred}$ of $N_p$ 3-tuples $< \tilde{b}^{sub}, \hat{p}, \tilde{b}^{obj} >$, which is fed into the entity refinement module $\mathcal{C}$.

**Conditioned Entity Refinement** $\mathcal{C}$   While not accurate, the bounding boxes $\{\tilde{b}^{sub}, \tilde{b}^{obj}\}$ received from $\mathcal{H}$, provide an initial estimate of entity locations. The entity refinement model, $\mathcal{C}$, leverages these estimates to propose a set of $N_{ce}$ refined bounding boxes $\{\hat{b}^{sub}, \hat{b}^{obj}\}$, conditioned on the predicate label estimate $\hat{p}$. This is implemented with a second decoder that shares the features $\mathbf{F}_p$ produced by the feature extractor $\mathcal{F}$ and a set of queries designed to induce the conditional operation.

For each bounding box estimate in $\mathcal{S}_{pred}$, $\mathcal{C}$ predicts $N_{ce}$ refined entity bounding boxes. The queries used to generate these refinements are

$$Q_{ce}^{(i)} = Emb_b(\tilde{b}) + Emb_p(\hat{p}) + Q_0^{(i)} \text{ where } i \in \{1, 2, ..., N_{ce}\} ,\tag{9}$$

$Emb_b(.)$ and $Emb_p(.)$ are two embeddings and $\mathbf{Q}_0$ is a DETR-like randomly generated set of $N_{ce}$ queries, introduced to guarantee that the queries $Q_{ce}^{(i)}$ are distinct. From the conditional queries $\mathbf{Q}_{ce}$, $\mathcal{C}$ produces $N_{ce}$ distinct corrections by computing representations $\mathbf{Z}_{ce} = Attn_{stack}(\mathbf{Q}_{ce}, \mathbf{Q}_{ce}, \mathbf{F}_p) \in \mathcal{R}^{N_{ce} \times d}$. It follows from (1)-(4) that the queries $\mathbf{Q}_{ce}$ bias the attention of $\mathcal{C}$ to the bounding boxes $\tilde{b}^{sub}$ and $\tilde{b}^{obj}$. This constraint encourages $\mathcal{C}$ to predict boxes in a substantially smaller region, thereby *limiting the box search space*. Furthermore, because attention is modulated by the predicted predicate label $\hat{p}$, this search is selective for boxes that comply with the former, which encourages agreement between entities and predicates, thus limiting the entity label search space.

For each bounding box in $\mathcal{S}_{pred}$, this process produces $N_{ce}$ entity refinements, for a total of $N_{ce}^2$ 5-tuples. $k$ 5-tuples are chosen per predicate in $\mathcal{S}_{pred}$ to be part of the predicted scene graph $\mathcal{G}$, which contains $k \times N_p$ scene graph nodes.

**Training**   TraCQ is trained with a set prediction loss for triplet detection that generalizes the entity detection set prediction loss of (7). Denote by $\hat{y}_{\mathcal{H}} =< \tilde{b}^{sub}, \hat{p}, \tilde{b}^{obj} >$ the prediction of $\mathcal{H}$, by $y_{\mathcal{H}} =< b^{sub}, p, b^{obj} >$ its groundtruth, by $\hat{y}_{\mathcal{C}} =< \hat{y}^e, \hat{b}^e >$ the entity correction of $\mathcal{C}$ and by $y_{\mathcal{C}} =< y^e, b^e >$ its groundtruth. TraCQ predicts $N_p$ relationships, where $N_p$ is larger than the number of relations in any given image. Similarly to DETR, this is handled by padding the ground truth set of relations with *no-relation* tokens $\phi$. This circumvents a difficulty of single-stage models, which must account for the fact that relations of the types < valid subj - no rel - valid object> are different from those of the type < no subj - no rel - no object >, or other combinations. The predicate detection matching cost $\mathcal{C}_{match}(\hat{y}_{\mathcal{H}}, y_{\mathcal{H}})$ between a predicted and a ground truth triplet generalizes the matching cost function of (6), considering both the predicate class prediction and the similarity of predicted $\tilde{b}$ and ground truth $b$ subject/object boxes, according to

$$\mathcal{C}_{match}(\hat{y}_{\mathcal{H}}, y_{\mathcal{H}}) = -\mathbb{1}_{\{p \neq \phi\}} \mathbf{p}_{y_{\mathcal{H}}}^p + \mathbb{1}_{\{p \neq \phi\}}[\mathcal{L}_{box}(\tilde{b}^{sub}, b^{sub}) + \mathcal{L}_{box}(\tilde{b}^{obj}, b^{obj})]\tag{10}$$

Similarly, the entity refinement matching cost $\mathcal{C}_{match}(\hat{y}_{\mathcal{C}}, y_{\mathcal{C}})$ uses both entity boxes and labels to calculate similarity, according to

$$\mathcal{C}_{match}(\hat{y}_{\mathcal{C}}, y_{\mathcal{C}}) = -\mathbb{1}_{\{y^e \neq \phi\}} \mathbf{p}_{y_{\mathcal{C}}}^{y^e} + \mathbb{1}_{\{y^e \neq \phi\}} \mathcal{L}_{box}(\hat{b}^e, b^e)\tag{11}$$

where the groundtruth entity $e$ is either the subject ($sub$) or object ($obj$) entity of the corresponding prediction by $\mathcal{H}$. For both subject and object, the box alignment cost is

$$\mathcal{L}_{box}(\hat{b}, b) = \lambda_{gIoU} \mathcal{L}_{gIoU}(\hat{b}, b) + \lambda_{L_1} \mathcal{L}_{L_1}(\hat{b}, b)$$

Given the triplet cost matrix $\mathbf{C}_{match}$, the Hungarian algorithm [19] is executed for the bipartite matching and each ground truth triplet $i$ is assigned to a predicted triplet $\hat{\sigma}(i)$. Let the $\hat{\sigma}_{\mathcal{H}}$ be the matching of $\mathcal{H}$ and $\hat{\sigma}_{\mathcal{C}}$ that of $\mathcal{C}$. Two losses are then defined as

$$\mathcal{L}_p = \sum_{i=1}^{N_p} \left[ \lambda_{lbl} \mathcal{L}_{cls}(\hat{p}_j, p_i) + \mathbb{1}_{\{p_i \neq \phi\}} \{\mathcal{L}_{box}(\tilde{b}_j^{sub}, b_i^{sub}) + \mathcal{L}_{box}(\tilde{b}_j^{obj}, b_i^{obj})\} \right] \Big|_{j=\hat{\sigma}_{\mathcal{H}}(i)}\tag{12}$$

$$\mathcal{L}_e = \sum_{i=1}^{N_p} \mathbb{1}_{\{p_i \neq \phi\}} \left\{ \sum_{j=1}^{N_{ce}} [\lambda_{lbl} \mathcal{L}_{cls}(\hat{y}_k^e, y_j^e) + \mathbb{1}_{\{y_j^e \neq \phi\}} \mathcal{L}_{box}(\hat{b}_k^e, b_j^e)] \Big|_{k=\hat{\sigma}_{\mathcal{C}}(j)} \right\}\tag{13}$$

where $\mathcal{L}_{cls}$ is the cross-entropy loss. $\mathcal{F}$ and $\mathcal{H}$ are trained using $\mathcal{L}_p$, whereas $\mathcal{C}$ is trained with $\mathcal{L}_e$. This encourages the decoupling of feature spaces, encouraging the features of $\mathcal{F}$ and $\mathcal{H}$ to specialize in predicate prediction and those of $\mathcal{C}$ on predicate-conditioned entity detection.

**Inference**    The inference stage involves combining the outputs of entity refinement FFNs of $\mathcal{C}$ and the predicate detection FFNs of $\mathcal{H}$ to form final triplets. Due to the conditional structure of the decoder, the predicate detection outputs and the set of $k$ entity refinement outputs have a one-to-one correspondence. Therefore, $k \times N_p$ SPO-tuples $< (b^{sub}, y^{sub}) - p - (b^{obj}, y^{obj}) >$ are automatically generated.

An SGG triplet score $s_{triplet}$ is computed per SPO-tuple with $s_{triplet} = s_{pred}s_{sub}s_{obj}$ where $s_{sub}$ and $s_{obj}$ are the scores of subject and object classification from $\mathcal{C}$, respectively, and $s_{pred}$ is the predicate classification score from $\mathcal{H}$. Finally, the predictions are sorted by descending $s_{triplet}$ and the top $m < kN_p$ predictions selected as nodes of the predicted scene graph $\mathcal{G}$.

**Comparison to previous approaches**    When compared to two-stage SGG models, TraCQ replaces the model of (8) by

$$Pr(\mathcal{G}|\mathcal{I}) = \sum_{\hat{\mathcal{B}}} Pr(\mathcal{P}, \hat{\mathcal{B}}|\mathcal{I})Pr(\mathcal{Y}, \mathcal{B}|\mathcal{P}, \hat{\mathcal{B}}, \mathcal{I}), \tag{14}$$

where $\hat{\mathcal{B}}$ is the set of bounding box estimates computed by $\mathcal{H}$ and the marginalization over $\hat{\mathcal{B}}$ is performed by the conditioning of the queries of $\mathcal{C}$ on the predictions of $\mathcal{H}$. Assume, for simplicity, that the labels $\mathcal{Y}, \mathcal{P}$ are independent of the bounding boxes $\mathcal{B}$. Then, (8) reduces to $Pr(\mathcal{G}|\mathcal{I}) = Pr(\mathcal{Y}|\mathcal{I})Pr(\mathcal{B}|\mathcal{I})Pr(\mathcal{P}|\mathcal{Y}, \mathcal{I})$ and its modeling complexity is dominated by the term $Pr(\mathcal{P}|\mathcal{Y}, \mathcal{I})$, which is a distribution conditioned the large combinatorial space of $O(\mathcal{E} \times \mathcal{E})$ entity pairs. On the other hand, (14) reduces to $Pr(\mathcal{G}|\mathcal{I}) = Pr(\mathcal{P}|\mathcal{I})Pr(\mathcal{B}|\mathcal{I})Pr(\mathcal{Y}|\mathcal{P}, \mathcal{I})$ and its complexity is determined by $Pr(\mathcal{Y}|\mathcal{P}, \mathcal{I})$, which is a distribution conditioned on the non-combinatorial space of predicates. Hence, TraCQ has an innate advantage over the two-stage models, the modeling of probabilities conditioned on much smaller spaces, which allows it to perform inference much more efficiently. Furthermore, the conditional queries $\mathbf{Q}_{ce}$ enforce something akin to ROI pooling, highly constraining the refinements made by $\mathcal{C}$. First, these are constrained to the region of the bounding boxes predicted by $\mathcal{H}$. Second, the queries are generated with the predicate label as well and therefore must learn only those entities that are consistent with this label.

When compared to single-stage models, TraCQ has the advantage of a feature representation where entities and predicates are less entangled, due to the training of the two decoders with different loss functions, and the use of decoder $\mathcal{C}$ to predict the entity labels and refine the bounding boxes predicted by $\mathcal{H}$. This places less stress on the training of $\mathcal{F}$ and $\mathcal{H}$ to produce high quality features for entity localization and encourages $\mathcal{C}$ to produce such features.

In other one-stage models, which predict combinations of entities and decouple just the predicate classification from entity classification, we observe that the features are not fully disentangled leading to drop in performance. The disentanglement induced by the predicate detection task reduces the feature representation space needed to detect the predicates and allows the use of a lightweight model to learn it.

## 5    Experiments

In this section, we present the results of TraCQ and ablates its components.

### 5.1    Settings

**Dataset**. Visual Genome (VG) [18] is a popular benchmark for SGG. While it is a large dataset containing 75k object categories and 37k predicate categories, both object and predicate distributions are highly long-tailed, where most of the categories only have few instances. Hence, a popular subset VG150 of VG is proposed by [43], which contains the most frequent 150 object classes and 50 predicate classes. We follow the setting of prior works, adopting VG150 in all the experiments.

**Metrics**. Since the annotations for SGG datasets are usually incomplete, early works in SGG are mostly evaluated with Recall@K (R@K). However, since SGG is a highly long-tailed problem, most recent research [4, 34] also present mean Recall@K (R@K), $K = \{20, 50, 100\}$ as the metric.

Table 2: Quantitative results of TraCQ in comparison with state-of-the-art methods on the VG dataset. TraCQ achieves new state-of-the-art results outperforming existing one-stage and two-stage models in most metrics, while reducing model complexity without the need for any extra features (e.g., glove vector, knowledge graph, etc.). Note that '-' indicates that the corresponding results are unavailable.

| | Method | Extra Features | Backbone | mean-Recall (↑) @20 | @50 | @100 | Recall (↑) @20 | @50 | @100 | #params (↓) (M) |
|---|---|---|---|---|---|---|---|---|---|---|
| Two-Stage | MOTIFS[52] | ✓ | X-101FPN | 4.2 | 5.7 | 6.6 | 21.4 | 27.2 | 30.5 | 240.7 |
| | KERN[4] | ✓ | VGG16 | - | 6.4 | 7.3 | 22.3 | 27.1 | - | 405.2 |
| | GPS-Net[25] | ✓ | VGG16 | 6.9 | 8.7 | 9.8 | 22.3 | 28.9 | 33.2 | - |
| | BGNN[21] | ✓ | X-101FPN | 7.5 | 10.7 | 13.6 | 23.3 | 31.0 | 34.6 | 341.9 |
| | VCTree-TDE[34] | ✓ | X-101FPN | 6.3 | 9.3 | 11.1 | 14.3 | 19.6 | 23.2 | 360.8 |
| | IMP+[43] | ✗ | VGG16 | 2.9 | 3.8 | 4.8 | 14.6 | 20.7 | 24.5 | 203.8 |
| | G-RCNN[45] | ✗ | VGG16 | - | - | - | - | 11.4 | 13.7 | - |
| One Stage | FCSGG[26] | ✗ | HRNetW48-5S-FPN | 2.7 | 3.6 | 4.2 | 16.1 | 21.3 | 25.1 | 87.1 |
| | RelTR[5] | ✗ | ResNet-50 | 5.8 | 8.5 | - | 20.2 | 25.2 | - | 63.7 |
| | Relationformer[33] | ✗ | ResNet-50 | 4.6 | 9.3 | 10.7 | 22.2 | 28.4 | 31.3 | 92.9 |
| | **TraCQ (ours)** | ✗ | ResNet-50 | **12.0** | **13.8** | **14.6** | 19.7 | 28.3 | **35.7** | **51.2** |

## 5.2 Implementation details

The PnP-DETR[39] is set with sample ratio $\alpha$ of 0.33 and pool samples $M$ of 60. We adopt ResNet-50 [10] with a 6-layer transformer encoder as the visual feature extractor $\mathcal{F}$. Both decoders $\mathcal{H}$ and $\mathcal{C}$ have a 6-layer transformer, with 8 and 4 attention heads respectively. The number of queries $N_p$ is set to 200 and $N_{ce}$ is set to 10 with $k = 5$. The FFNs for bounding box prediction have 3 linear layers with ReLU, while the FFNs for predicting object and relation labels have one linear layer.

We initialize the network with the parameters of PnP-DETR trained on VG for object detection, and adopt the default weight coefficients $\lambda_{L_1}$, $\lambda_{GIoU}$ and $\lambda_{lbl}$ from [39]. The network is optimized with AdamW [27] where the weight decay is $10^{-4}$. The learning rates for training the backbone and the rest modules are $10^{-5}$ and $10^{-4}$ respectively. All experiments are conducted on 8 NVIDIA TITAN X GPUs, with total batch size of 32.

## 5.3 Comparisons to baselines

The baselines in Table 1 show the importance of disentanglement, where the performance increase from SD to DDTR. However, disjoint queries lead to large computational overhead due to the post-processing matching step and there is no way for further joint optimization. Learning from these baselines, TraCQ is a DD-type model leveraging decoupled training to disentangle the two distributions in representation learning while maintaining certain degree of dependence between the two tasks. As a result, TraCQ outperforms all these baselines with a large margin (2.8 point gain over DDTR in mR@20), while maintaining similar number of parameters as DD and achieving 1.5x faster inference than DDTR. This highlights that complete disentanglement is not possible, since the detection of a predicate always requires some knowledge of the associated subject and object.

## 5.4 Comparisons to prior methods

We compare scores of R@K, mR@K and number of parameters of TraCQ with several two-stage and one-stage models in Table 2. The usage of extra features such as semantic and statistic information by the model is indicated to distinguish those models from visual appearance based models. TraCQ beats all models in terms of mR@K while having 20% fewer parameters than [5], reportedly having the smallest model size. It also exhibits a much less drastic drop in the mean recall values when we go from 100 to 20 for mR@K. This shows that the model is more confident about it's predictions than others. In R@20 and R@50, TraCQ lags behind only those two-stage models, that are heavier and/or use extra features in their pipeline. TracQ is the first one-stage model that beats two-stage models in terms of performance with a much smaller model size.

## 5.5 Ablation studies

We perform the following ablations on TraCQ to understand the influence of each component.

Table 3: **Ablations on the orders** of entity detection and predicate detection.

| Model | mean Recall ($\uparrow$) | | |
|---|---|---|---|
| | @20 | @50 | @100 |
| Entity-first | 11.2 | 12.3 | 12.7 |
| Predicate-first (ours) | 12.0 | 13.8 | 14.6 |

Table 4: **Ablation on the hyper-parameter k**. SGG is evaluated with mR@20 and R@20.

| k | #Predictions ($\downarrow$) | mR@20 ($\uparrow$) | R@20 ($\uparrow$) |
|---|---|---|---|
| 1 | **200** | 8.9 | 18.4 |
| 5 | 1000 | 12.0 | **19.7** |
| 10 | 2000 | **12.1** | **19.7** |
| 15 | 3000 | **12.1** | **19.7** |

Table 5: **Ablation on bounding box refinements**.

| Entity box prediction $\hat{b}^e$ | mR@ 20 / 50 / 100 |
|---|---|
| From $\mathcal{H}$ (no correction) | 17.2 / 18.0 / 18.1 |
| From $\mathcal{C}$ (ours) | 16.9 / 18.6 / 19.2 |

Table 6: **Ablation on conditioned queries**.

| Conditioned queries | mR@ 20 / 50 / 100 |
|---|---|
| w/o $Emb_p(\hat{p})$ | 10.8 / 12.4 / 13.1 |
| $\mathcal{Q}_{ce}$ (ours) | 12.0 / 13.8 / 14.6 |

**Ablations on the formulation**   To validate the effectiveness of the proposed formulation, we also conduct the experiment of the entity-first variant. Different from TraCQ that predicts entity labels based on predicate predictions, this variant performs entity detection first and conditions predicate classification on it. We maintain similar setup as TraCQ on other components to ensure fair comparison. Table 3 shows that this variant underperforms the proposed formulation. This supports our idea of avoiding the combinatorial entity label space.

**Ablations on the hyperparameter k**   The hyperparameter k controls the number of candidate entity bounding boxes. We evaluate TraCQ with different values of k and present the mR@20 and R@20 in Table 4. We can see that the performance saturates quickly as the number of predictions (denoted as #Predictions) increases. Since the inference time is proportional to k, we pick $k = 5$.

**Ablations on bounding box refinements**   The entity refinement module, $\mathcal{C}$, performs entity detection using the global features and the conditioned queries of entity bounding boxes and predicates. To study how much box refinement was happening from $H$ to $\mathcal{C}$, we directly take the predicted bounding box of subject and object, i.e. $\tilde{b}^{sub}$ and $\tilde{b}^{obj}$, from the predicate decoder $\mathcal{H}$ as the final prediction, but not correct the entity bounding boxes with $\mathcal{C}$. Since $\mathcal{H}$ does not produce class labels for subjects and objects, we evaluate the network in the task of Predicate Detection (PredDet), where ground-truth entity labels are given and the task is to predict $< b^{sub} - p - b^{obj} >$ tuples. Table 5 presents such results compared to TraCQ under the mR@K metric, which shows that the model performance drops when removing the entity refinement module.

**Ablations on conditioned queries**   We also ablate the impact of using the predicate labels $\hat{p}$ for generating the predicate conditioned queries $\mathcal{Q}_{ce}$ for $\mathcal{C}$. We retrain $\mathcal{C}$ conditioned only on the bounding boxes from $\mathcal{H}$, namely, removing $Emb_p(\hat{p})$ from (9). The bounding box conditioning cannot be removed, since that reduces $\mathcal{C}$ to a general entity set detection model. Table 6 presents the effect of removing the predicate conditioning for $\mathcal{C}$. The performance of the overall model drops by $\sim 2$ points at mR@20. This shows that the explicit condition forced onto $\mathcal{C}$ by the predicate conditioned queries does in fact reduce the searching space for inferring entity labels.

## 6   Related work

**Scene graph generation (SGG)** has long been studied in the literature due to its wide applicability [48, 47, 13, 14, 38, 15, 22, 36, 15, 22, 36]. Since SGG is a highly complex problem, pioneer works [18, 43, 52] leverage marginalization to decompose the problem into predicate classification, entity classification and entity localization. This formulation largely impacts the latter research. Most of the works follow the same pipeline structure which first detects a set of entity proposals with state-of-the-art object detectors [7, 31] and then classify all the pairs of entity proposals into a predicate class at the second stage. While object detection is considered as solved, most research focuses on learning a better mapping between entity pairs and predicate classes [54, 35, 4, 50, 24, 8, 23, 40, 42]. This can be achieved either by learning better representations for contextual reasoning [52, 21, 4, 41, 42, 46, 49, 17, 28], leveraging external knowledge [9, 51, 8, 50], or unbiased learning strategy [34, 25, 40, 1, 44, 6]. Among these, Relation Transformer Network[17] and Lu et

al [28] are pioneering works adopting Transformer encoder-decoder pair for learning SGG, while both of them rely on bounding box predictions from a CNN-based object detector. These research has provided insightful observations and promising results. However, the two-stage formulation has a fundamental problem, the searching space for relation detection is always $\mathcal{O}(N^2)$, which can retard the inference speed. In addition, building another computation module on top of a large object detector is inefficient in terms of parameters and computations, which prevents end-to-end optimizations for the SGG task.

To address this, [26, 5, 33, 20] propose one-stage models for SGG. [26] encodes objects as center points and relationships as vector fields. Although it is lightweight and fast, it has a significant performance gap compared to other two-stage methods. Others adopt DETR [3]-based architectures. RelTR[5] combines subject and object queries to decode the triplet predictions, while [33] produces a set of object tokens and a relation token, and decodes them respectively. [20] is a concurrent work that requires bipartite graph assembling similar to DDTR. While these works provide inspirational ideas and allows end-to-end optimizations, they still perform poor than two-stage methods. We hypothesize that the reason is because of the entangled feature space of entities and predicates. This paper focuses on a single-stage model with less entanglement between the two distributions.

**Transformer-based Detection** has become popular for many computer vision tasks recently. The pioneering work, DETR [3], introduces the set prediction with a Transformer encoder-decoder architecture, where the encoder learns contextual features and the decoder takes object queries to generate object proposals. This drastically changes the way object detection is viewed, and inspires several later works on other tasks, such as instance segmentation [12], visual grounding [11], multi-object tracking [53] and HOI detection [16]. HOI detection is most similar to our task, SGG, in that it localizes and recognizes the relationships of each human-object pair. However, there are inherent differences between these two tasks. The HOI task does not need to deal with the $\mathcal{O}(\mathcal{E} \times \mathcal{E})$ space since the subject of each interaction tuple is always the human in the scene. Hence, most of the HOI models cannot be directly transferred to tackle the complexity of the SGG task.

## 7   Conclusions

Scene graph generation (SGG) deals with two different distributions, namely entities and predicates. Most methods rely on detecting all entities, followed by combining pairs of entites. Unlike such two-stage methods, we propose an end-to-end trainable framework for SGG using a Transformer architecture which decouples these two distributions effectively with conditional queries (TraCQ), leading to a small model size and an efficient inference mechanism. TraCQ significantly outperforms existing single-stage methods, even beating the performance of many state-of-the-art two-stage methods on Visual Genome benchmark.

## 8   Societal Impacts

SGG models have wide applicability, e.g. caption generation [48, 47], visual question answering [13]. The model itself is harmless and helpful for providing a compact description of the scene. However, there may be some privacy concerns when the model is utilized for monitoring people and their interactions. Note that no explicit recognition is involved in this framework.

## 9   Limitations

In this paper, we focus on developing a single-stage detector for visual relations. We extensively study different formulations of SGG models with different degrees of entanglement between the predicate and entity detection task. However, we didn't deliberately aimed at addressing the long-tail nature of the two distributions in this work. The unbiased training techniques like resampling or reweighting can be considered in the future work.

## Acknowledgments

This work was partially funded by NSF awards IIS1924937 and IIS-2041009, a gift from Amazon, a gift from Qualcomm, and NVIDIA GPU donations. We also acknowledge and thank the use of the Nautilus platform for some of the experiments discussed above.

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
