# Single-Stage Visual Relationship Learning using Conditional Queries

**Alakh Desai[1], Tz-Ying Wu[1], Subarna Tripathi[2], Nuno Vasconcelos[1]**
[1]University of California San Diego, USA
[2]Intel Labs, USA

## A   Visual Examples

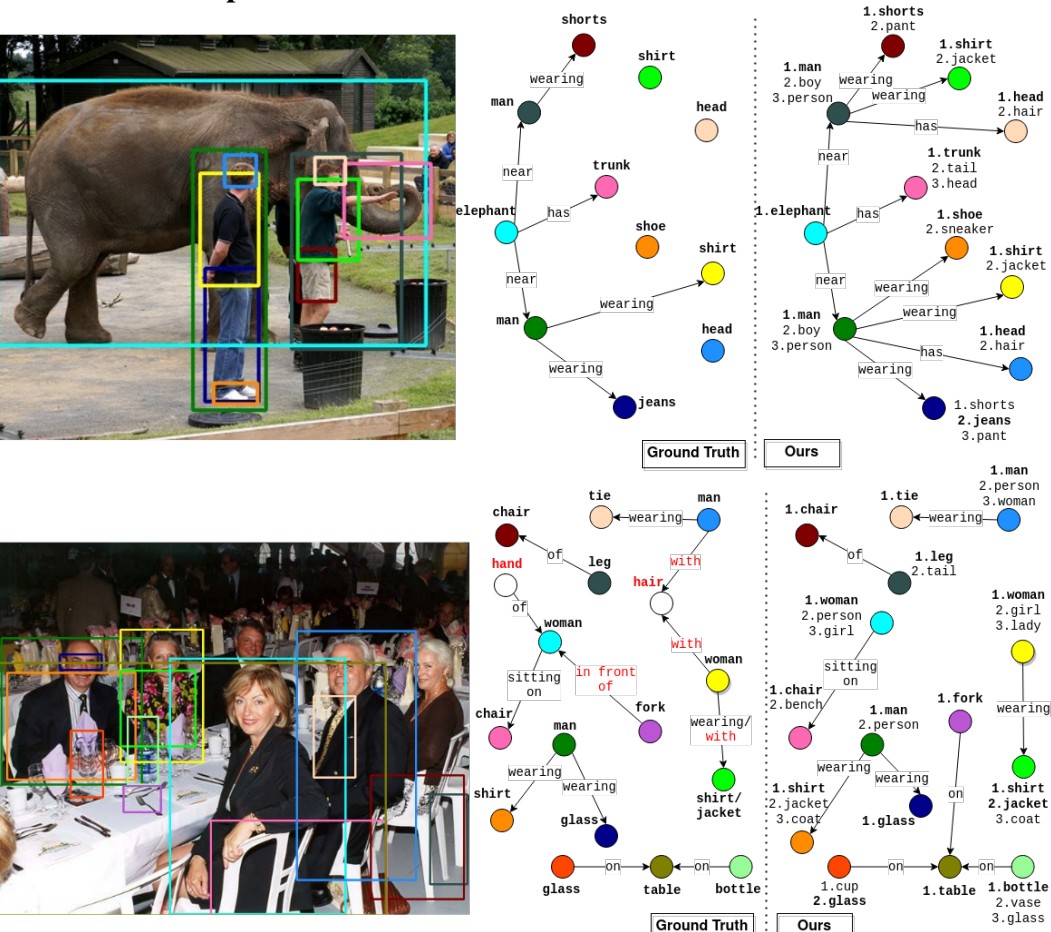

Figure 1: **Qualitative comparison in SGG**. In each sub-figure, colors of bounding boxes in the image (left) correspond to the entities (nodes) in the triplets. Text in red indicates incorrect/missed relation/entity.

Visual examples of SGG generated by TraCQ are presented in Figure 1. The examples show two things. First, TraCQ can predict reasonable relations that are missing in ground-truth annotations. For example, in the upper example, TraCQ predicts "wearing" as the predicate between "man" (dark green box) and "shoe" (orange box), while the annotation is missing in the ground truth. Second, most of the entity labels predicted by TraCQ are synonyms of the ground-truth labels, e.g. shorts/pants (dark red box) and shoe/sneaker (orange box) in the upper example, and woman/person (cyan box) in the lower example. These show that TraCQ can generate meaningful descriptions of the scene.

36th Conference on Neural Information Processing Systems (NeurIPS 2022).

# B  Details of baselines

As illustrated in Section 3.2 and Figure 1 of the main paper, SD and DD both employ a common set of random queries for entity and predicate predictions, while DDTR decouples the two tasks with separate sets of random queries, which requires further matching between the two sets of outputs. Suppose we have $M$ predicate and $N$ entity predictions, then, we get $M \times \mathbf{P}_2^N$ possible SPO tuple combinations. To find the best match between the two sets, we must build the cost matrix between each pair of these combinations, which is computationally expensive. This is visible in the inference time as shown in the Table 1 of the main paper. the contrary, TraCQ avoids the time-consuming bipartite matching by introducing conditional queries, such that a one-to-one mapping is maintained between the two sets of the outputs. This successfully reduces the inference time from 0.22s to 0.15s.

TraCQ and all the baselines (SD, DD, DDTR) generate equivalent numbers of predicate predictions (i.e. $N_p = 200$). However, TraCQ provides the flexibility to refine the entity predictions by sampling around the bounding box predictions from predicate decoder $\mathcal{H}$. Specifically, TraCQ generates $N_{ce}$ entity refinements for each bounding box prediction in $N_p$ predicate predictions, but chooses the top $k$ SPO-tuple candidates for each predicate (see Section 4 - Conditioned Entity Refinement in the main paper), which results in $k \times N_p$ predictions. To make it a fair comparison to TraCQ, we adopt the same top $k$ selection for DDTR with $k = 5$ as reported in Table 1 of the main paper. Compared to SD and DD, DDTR and TraCQ did generate more SPO-tuple predictions when $k = 5$. However, TraCQ outperform these two baselines, by a good margin, when $k = 1$ as well.

**Number of parameters**    As shown in Table 2 of the main paper, more parameters do not necessarily lead to better performance. Hence, the performance gain of TraCQ is not the direct outcome of a larger model. In fact, TraCQ is smaller than DDTR (51.2M vs 82.9M) but achieves better performance (mR@20: 12.0 vs 9.2). Since it is challenging to design a model of exact same size as the baselines, we ensure each sub-module of same type (Enc, Dec, FFN) shares the same number of parameters and the same hyper-parameters across baselines. The number of parameters for SD, DD, DDTR and TraCQ are 41.7M, 51.1M, 82.9M, 51.2M respectively.

**Number of predictions**    Generating more predictions does not guarantee higher Recall. Regardless of one-stage or two-stage models, the R@K of SGG models is evaluated by ranking all the generated SPO tuples for each image, but only the top K predictions are considered.

The model can only benefit from more predictions if the predictions are of high quality and the performance is not saturated. As shown in Table 4 of the main paper, the performance of TraCQ increases from $k = 1$ to $k = 5$, which shows that the entity refinement module generates high-quality refinements that correct the error of initial predictions. When $k > 5$, the performance saturates quickly and there is no benefit to generate more SPO-tuple candidates.

In fact, most prior works generate more predictions than TraCQ. For the models that predict predicates based on entity detection, there are $\mathbf{P}_2^N$ predictions, where $N$ is the number of bbox candidates. The Relationformer [1] mentioned by the reviewer ZNFu is such a case, which has number of predictions in $\mathcal{O}(N^2)$. However, DDTR and TraCQ only generate number of predictions in $\mathcal{O}(k \times N)$, where $k << N$.

# C  Refinement Module

Table 5 of the main paper presents the result of PredDet which is defined as a set prediction task of predicate nodes, that is, $< b_{sub} - p - b_{obj} >$ tuples. The input for this task is the image itself and the output is a set of 3-tuples. The idea is to keep the predicate label independent of the entity labels. This allows for better visual learning of the predicate labels, that is, the model learns to detect *playing* irrespective of whether a *child* or a *dog* is *playing*. This separation allows for better learning of $f_p$ and $f_i$.

The entity refinement module $\mathcal{C}$ not only refines the bbox predictions but also predicts the entity labels. $\mathcal{C}$ implicitly performs dense sampling around the bbox predictions from $\mathcal{H}$, which may not have visible impact on bbox refinement when the initial bbox prediction is precise, so the gain is relatively small for PredDet. However, $\mathcal{C}$ is important for learning the entity label distributions and prevents the entanglement of entity feature and predicate feature (see Sec. 3.2 of the main paper).

If we remove $\mathcal{C}$ and add a classification head to the predicate decoder $\mathcal{H}$, the model reduces to SD, which underperforms TraCQ. We also evaluate a variant, where $\mathcal{C}$ predicts only entity labels. The mR@20/50/100 of the suggested baseline and TraCQ are 10.7/11.7/12.1 and 12.0/13.8/14.6, respectively. This indicates that entity bboxes and labels are closely related, and optimizing for both training objectives helps the entity refinement module learn better.

**Visualization of entity box refinement** We also visualize the effect of entity bounding box refinement in Figure 2. We can see that the refined bounding boxes are tighter estimates for the subjects and objects in the scene.

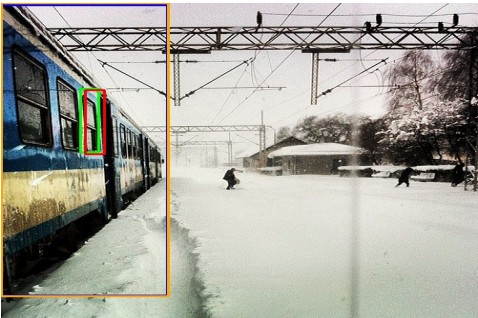 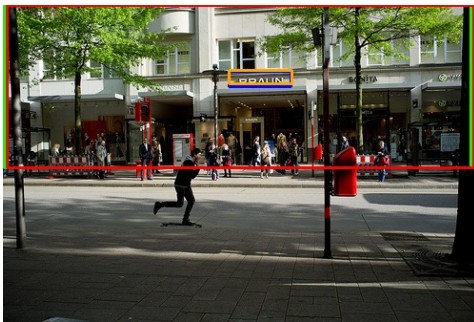

Figure 2: **Visualization of the effect of Refinement Module**. Different colored bounding boxes for subject/object are used to distinguish between the original bounding box (red/orange) and refined bounding box (green/blue). Left: *train-has-window*. Right: *sign-on-building*. (best viewed in color)

# References

[1] Suprosanna Shit, Rajat Koner, Bastian Wittmann, Johannes Paetzold, Ivan Ezhov, Hongwei Li, Jiazhen Pan, Sahand Sharifzadeh, Georgios Kaissis, Volker Tresp, and Bjoern Menze. Relationformer: A unified framework for image-to-graph generation, 2022.