# OpenReview forum: "Single-Stage Visual Relationship Learning using Conditional Queries"
_NeurIPS.cc/2022/Conference — NeurIPS 2022 Accept_

### Official Review · Reviewer_zyLi · 2022-07-05

**Rating:** 6
**Confidence:** 4
**Soundness:** 3 good
**Presentation:** 2 fair
**Contribution:** 2 fair

**Summary:**

The paper proposes a single-stage end-to-end trainable DETR-based architecture named TraCQ. It uses a separate encoder-decoder architecture that disentangles entity and predicate space. Unline earlier architecture they demonstrated strong coupling between entity space and predicate space hurts the performance. Hence it proposes conditional learning of predicate couple learning of entity scene and predicate prediction and establishes a weak coupling by loosely estimating bounding boxes of subject and object. This conditional and couple learning also reduces the parameter significantly. They conducted their experiment on Visual Genome Dataset and achieved a new benchmark on mean SGDet and SGDet recall.

**Questions:**

1. Have you conducted any experiments from class to predicate? as this is the normal order
2. Can you scale up for a large number of objects as the complexity is high?
Some of the suggestions:
Please add the object detection score and FPS in Tab.2 for completeness.


Good to have but doesn't impact the rating :
The presentation and writing can also be improved.
It is better to include some of the early transformer-based two-stage networks like :
1.Relation Transformer Network, Koner at el,
2. Context-aware Scene Graph Generation with Seq2Seq Transformers , Lu at el

**Limitations:**

The authors should provide some of the limitations of the method like:  is it applicable to other relation detection metrics (e.g. SGCls)

**Strengths And Weaknesses:**

Strength :
The idea has potential, disentanglement of predicate and entity space seems beneficial for predicate prediction. But there are some drawbacks which are mentioned later.


Weakness :
There are a few weaknesses in the paper:
1. It uses pnp DETR, but never mentioned its object detection performance, whereas SGDet is highly reliant on Object Detection performance. Despite the good performance on SGDet Table 3 shows the model is underperforming on Predicate Detection. It may possibly the performance may come from superior object detection performance. Authors should provide additional experiments.
2. What is the object-scene interaction feature $f_{i}$? its better to introduce or explain a bit.
3. Visual Genome has high biases, for a strong result section authors should test TraCQ on multiple datasets.

---

> ### Author Response · Authors · 2022-08-02
> **Thanks for your comment**
>
> **Please read the General comments for all reviewers.**
>
> Thanks for providing thoughtful comments. Minor issues will be corrected as suggested. All results and discussions will be added to the final version.
>
> **Performance of PnP-DETR**
>
> Please refer to the General comments -- Contributions of PnP-DETR.
>
> **What is the object-scene interaction feature $f_i$?**
>
> Please refer to the General comments -- Explanations of $f_i$.
>
> **From class to predicate?**
>
> In L234-L258 of the main paper, we discuss the benefits of the proposed formulation from probability perspective. Compared to $Pr(\mathcal{P}|\mathcal{Y}, \mathcal{I})$, $Pr(\mathcal{Y}|\mathcal{P}, \mathcal{I})$ prevents the combinatorial space of $\mathcal{O}(\mathcal{E}\times \mathcal{E})$. In addition to the theoretical analysis, we conduct the experiment of the suggested formulation (from class to predicate). However, this formulation underperforms TraCQ. The mR@20/50/100 of this variant and TraCQ are 11.2/12.3/12.7 and 12.0/13.8/14.6 respectively.
>
> **Scale up for a large number of objects?**
>
> Under the formulation of TraCQ, the number of predictions is bounded by \#predicate-queries (i.e. $N_p$) instead of \#entity-queries. Compared to prior methods that create all the combinations of entity pairs, TraCQ is easier to scale up for larger number of objects. Note that TraCQ can already support the maximum number of object instances in a scene of Visual Genome (i.e. 60).
>
> **Early transformer-based two-stage networks**
>
> Thanks for introducing these papers. Relation Transformer Network and Lu et al. are both two-stage SGG models. They leverage a CNN-based object detector to generate entity bbox predictions, followed by a Transformer encoder-decoder pair to learn better context representations. We will add the citations and discussions of these papers.
>
> **Is it applicable to other relation detection metrics (e.g. SGCls)?**
>
> Evaluation metrics like PredCls, SGCls are mainly the results of the formulation of $Pr(\mathcal{B}|\mathcal{I})Pr(\mathcal{Y}|\mathcal{B}, \mathcal{I})Pr(\mathcal{P}|\mathcal{B}, \mathcal{Y}, \mathcal{I})$, which usually takes bbox prediction as the first step. However, these metrics do not make sense to most single-stage SGG models, since the ground truth information cannot be given directly. On the other hand, TraCQ and its variants can be evaluated on another related task, PredDet, that takes object classes (a set of object-level tags) as input and outputs visual relation triplets grounded on the image. For more details of PredDet, please refer to General comments -- Effectiveness of the refinement module.

---

> ### Comment · Reviewer_zyLi · 2022-08-09
> **Response to Rebuttal**
>
> I have read the response of the reviewer. it solves my most of concerns. The paper has technical novelty. I will change my rating to weak accept.

---

### Official Review · Reviewer_zNec · 2022-07-11

**Rating:** 5
**Confidence:** 3
**Soundness:** 3 good
**Presentation:** 3 good
**Contribution:** 3 good

**Summary:**

This work proposes an end-to-end trainable framework for SGG using a Transformer architecture that models entity and predicate distributions with conditional queries simultaneously, and then weakly couples predicate and entity detection through the refinement of entity locations & the prediction of labels. It achieves a decent performance on the Visual Genome benchmark.

**Questions:**

Overall this paper gives a clear introduction to their motivation and solutions. One minor question is the preliminaries. I'm not sure how Dual decoders actually work, and the detailed experimental setup of SS & DD.

**Limitations:**

The author already mentioned the limitation of their work in Suppl, which is their method doesn't take into account long-tail distribution into consideration. I expect to see further improvement in the future to address these issues.

**Strengths And Weaknesses:**

Strengths:
This paper is well organized, clearly written, and easy to follow. It has a clear explanation of the motivation based on the observation of the preliminary works. Besides, I like the design that decouples entity and predicate distributions with conditional queries and couples them with another instance refinement decoder. The final experiment results also support their claim and show superior performance in the widely-used datasets.

Weaknesses:
1. Missing literature: The authors do not include some recent works, like papers on SGG [1] and the similar human-object interaction detection task [2].
2. I'm not sure whether this work really achieves SOTA as it claims as I find the results in [1] are higher (but they use RN101, maybe an unfair comparison).
3. Typo: Line 135, two 'use of'

[1] Li et.al. SGTR: End-to-end Scene Graph Generation with Transformer
[2] Kim et.al. Hotr: End-to-end human-object interaction detection with transformers

---

> ### Author Response · Authors · 2022-08-02
> **Thanks for your comment**
>
> **Please read the General comments for all reviewers.**
>
> Thanks for providing thoughtful comments. Minor issues will be corrected as suggested. All results and discussions will be added to the final version.
>
> **Missing literature**
>
> Human-object-interaction detection (HOI) is a task similar to the SGG task. However, there are inherent differences between these two tasks. The HOI task does not need to deal with the $\mathcal{O}(\mathcal{E}\times \mathcal{E})$ space since the subject of each interaction tuple is always the human in the scene. Hence, most of the HOI models cannot be directly transferred to tackle the complexity of the SGG task.
>
> Thank you for pointing to SGTR. SGTR is a concurrent work that was published after this submission. The mR@50/100 of TraCQ and SGTR are comparable (13.8/14.6 vs 12.0/15.2) without the unbiased resampling strategy (bi-level sampling), while TraCQ achieves superior performance in R@50/100 (28.3/35.7 vs 24.6/28.4). Note that SGTR adopts a deeper backbone (ResNet-101) than TraCQ (ResNet-50). We will add the citations and discussions of these papers.
>
> **Details of the baselines**
>
> Dual decoder (DD) employs a common set of random queries for predicate prediction and entity prediction, and thus the outputs from the two decoders inherently have one-on-one mapping. We provide more details of the baselines in the General comments.

---

### Official Review · Reviewer_ZNFu · 2022-07-11

**Rating:** 4
**Confidence:** 4
**Soundness:** 2 fair
**Presentation:** 3 good
**Contribution:** 2 fair

**Summary:**

The authors tackle the task of scene graph generation by proposing a new formulation which models the problem as a product of two conditionals - the first of which generates a bounding box estimate and a predicate label, and the second takes the aforementioned estimates to get subject and object class labels and refined bounding boxes. The two distributions are realized via a DETR style encoder-decoder transformer architecture. The experimental analysis shows improved performance compared to the baselines on the Visual Genome dataset.

**Questions:**

Although I like the novel formulation resulting in a unique factorization of the SGG problem (Equation 14), there are certain issues with experiments and the described analysis that makes it harder for me to evaluate the proposed approach.

1. It would be ideal if the authors could further clarify the analysis in Table 1. More specifically, what are the hyperparameters used by the different baseline models and whether they have the same number of parameters. If they don't, why is the performance improvement from SD to DD to DDTR not a direct consequence of improved model capacity. Additionally, when comparing TraCQ to the baselines, the performance under the same number of predictions should be shown. (See points 2 and 3 in weaknesses)

2. The authors should clarify the importance of the refinement module $\mathcal{C}$. Why is the refinement module $\mathcal{C}$ necessary? Why is the predicate decoder $\mathcal{H}$ unable to predict entity labels, when it already predicts a reasonable estimate of the entity bounding boxes? (See point 4 in weaknesses)

3. Using pretrained weights adds certain biases to the predicate decoder $\mathcal{H}$. Would initializing the weights randomly make the learning much harder ? (See point 5 in weaknesses)

4. The authors should report ablations on parameters $N_{ce}, N_p$ and also report the zero shot recall performance. Additionally, they should also report the per class predicate performance to help understand what predicate classes the model does better on (See point 6 in weaknesses)

**Limitations:**

The authors have addressed certain limitations and potential negative societal impact of their work. Other limitations could arise from the choice of DETR architecture. For example, the vanilla DETR model ([3]) has poorer performance on detecting smaller objects (see [b]).

[b] Zhu et al. "Deformable DETR: Deformable Transformers for End-to-End Object Detection." ICLR 2021.

**Strengths And Weaknesses:**

$$\large{\textbf{Strengths}}$$

1. The paper is well written and easy to follow. The motivation is well explained and Figure 2 provides a clear overview of the approach.
2. The factorization assumed by the paper (Equation 14) is novel compared to existing work which often condition the predicate prediction on the subject/objects (Equation 8). This formulation provides interesting opportunities for future research.

$$\large{\textbf{Weaknesses}}$$

1. The claims learning in the non-combinatorial predicate space is easier than the $O(\varepsilon \times \varepsilon)$ entity pair space due to the former being smaller. This is supported by analysis in Section 4 (Comparision to Previous Approaches subsection). Firstly, the analysis makes the assumption that labels $\mathcal{Y}, \mathcal{P}$ are independent of bounding boxes $\mathcal{B}$ to simplify the factorization in Equation 14. This assumption clearly doesn't hold in the context of scene graph generation as bounding boxes are crucial to localizing where in the image a particular interaction is happening, and between what objects the aforementioned interaction occurs. Therefore, the simplification of the factorization in Equation 14 under this assumption is misleading. Secondly, looking at Equation 14 without the simplification, it is unclear why the term $\Pr(\mathcal{Y}, \mathcal{B} | \mathcal{P}, \hat{\mathcal{B}}, \mathcal{I})$ should be better to learn from. Even though it is conditioned on a smaller space as compared to the term $\Pr(\mathcal{P}| \mathcal{B}, \mathcal{Y}, \mathcal{I})$ in Equation 8, the proposed formulation is still predicting in a much larger space consisting of entities labels $\mathcal{Y}$ and bounding boxes $ \mathcal{B}$  (which are continuous in nature).

2. The analysis in Table 1 shows the model performance to improve when increasing the amount of disentanglement. However, this performance improvement could be a direct result of increasing the model capacity. For example, going from SD to DD would entail an additional decoder (which I'm assuming is 6 layers deep based on the experiments), which could provide the observed gains as the model has a greater capacity to fit the training data. As the authors haven't discussed the hyperparameters for the three models shown, disentanglement cannot solely be considered as the cause for the observed improvements.

3. Building on the previous point, Table 2 in the supplementary (ablation on the hyperparameter k) shows a considerable improvement in performance when going from k=1 to k=5, as the number of predictions generated by the model increasing from 200 to 1000. The reported performance gains of the proposed TraCQ to the baselines in Table 1 of the supplementary, and to other single stage transformer models (like [25]) in Table 2 of the main paper are misleading due to the increased number of predictions made by the model. This is majorly due to the recall metrics not checking for prediction precision. The authors have not mentioned the number of queries used by the models in Table 1. If, for example, it is 200, then the proposed TraCQ model would be worse than the DDTR model on mR@20 assuming the same number of predictions (200).

4. From the ablation in Table 3, the refinement module $\mathcal{C}$ doesn't impact the overall model performance as much. Looking at the numbers, it reduces performance on mR@20, which is a stricter and more desirable metric compared to mR@50/100. The visualizations in Figure 2 of the supplementary does not show the change in the bounding box before and after refinement, making it difficult to analyze the importance of the proposed module $\mathcal{C}$. As the module $\mathcal{H}$ already predicts the predicate class and provides a reasonable estimate of the object bounding boxes, one could easily add another couple detection heads to $\mathcal{H}$ to predict the entity class labels as well. This would do away with the entity refinement module (which has almost the same number of parameters as $\mathcal{H}$). The authors should provide an ablation / address why this simpler architecture is worse than the proposed TraCQ.

5. The authors mention that the initialize the network with parameters pre-trained on VG object detection task. Doing so already biases the predicate decoder $\mathcal{H}$ to be aware of the entity labels (as the object detector was trained to predict both bounding boxes and labels). It is unclear what effect this initialization has on the overall performance, and whether learning $\mathcal{H}$ would be much harder if the weights were initialized randomly.

6. There are some missing ablations with regards to the what effect do the parameters $N_{ce}, N_p$ have on the performance. Additionally, the authors should also report the zero-shot recall performance (introduced in [a]) that computes the recall@K for triplets not present in the training data. It would highlight the generalizability of the proposed approach.

[a] Lu, Cewu, et al. "Visual relationship detection with language priors." European conference on computer vision. Springer, Cham, 2016.

---

> ### Author Response · Authors · 2022-08-02
> **Thanks for your comment**
>
> **Please read the General comments for all reviewers.**
>
> Thanks for providing thoughtful comments. Minor issues will be corrected as suggested. All results and discussions will be added to the final version.
>
> **Why this formulation?**
>
> The analysis in L234-L258 of the main paper is just to provide an intuition of the proposed formulation. To make the idea easier to follow, we simplify the probability with the assumption of independence. This is like most papers that assume the data distributions are Gaussian distributions. Here we give more intuitions about why we learn entity classes given predicates (predicate-first) rather than learning predicates given entities (entity-first). It is true that the latter is easier, but the underlying assumption is that the entity pairs are known. Which is harder in general. This is hinted by the fact that, there is a significant performance drop when moving from PredCls task to SGCls task. For example, the mR@20 of MOTIFS for PredCls/SGCls is 10.8/6.3. We also conduct the experiment of the entity-first variant, and empirically show that this formulation underperforms the predicate-first formulation in TraCQ. The mR@20/50/100 of this variant and TraCQ are 11.2/12.3/12.7 and 12.0/13.8/14.6 respectively.
>
> To put it from the motivational point of view, we humans can always perceive an object being "held” in a picture, even if the object category is unknown. Visual features around and including the entity pairs play the most important role for the predicate classification, not the exact categories of the entities. Over-dependence on entity classes may lead the predicate classifier to learn a shortcut from the NLP bias. For example, for a detected pair <person, blanket>, most SGG models will be highly biased toward predicting "covered in" predicate, whereas in reality, the triplet could be the "person *folding* a blanket". Additionally, as discussed in section 3.2 of the main paper, learning two different long-tailed classifiers, namely entity and predicate, together is non-trivial. Hence, we take the approach of disentanglement, where two separate decoders are employed to model the two distinct distributions.
>
> **Clarifications about the baseline models**
>
> Please refer to the General comments --  Details of baselines, Number of predictions, Number of parameters.
>
> **Why is the refinement module important?**
>
> Please refer to the General comments -- Effectiveness of the refinement module.
>
> **Initialization biased for better box detection**
>
> Only the ResNet-50 backbone is initialized by the pretrained model, but not the decoder/Query/FFNs, which are the most important part of DETR. Hence, the predicate decoder is not biased by the pretrained weights.

---

> > ### Comment · Reviewer_ZNFu · 2022-08-09
> > **Response to Rebuttal**
> >
> > The authors have addressed some of my initial concerns. Therefore, I'm changing my rating based on the rebuttal. I still, however, have some concerns.
> >
> > 1. I disagree with the authors claim that generating more predictions doesn't lead to improved performance. The authors are correct when they say that mR@K and R@K consider only consider the top K predictions, therefore requiring the model to accurately rank them. However, generating $k=5$ predictions for each estimated predicate allows the model to hedge its bets in certain cases. For example, consider a situation where for a given predicate and initial bbox estimates by $H$, the model is unsure about the correct class label wherein multiple object classes have similar prediction scores (for example similar classes like tower and building). Typically, you would just take an argmax, therefore making a hard assignment in a situation where multiple classes are possibly viable. If the result of the argmax does not match the ground truth, your recall metric will count that as an incorrect prediction. However, when you choose $k=5$, you can get multiple viable predictions (one of which is correct) into your top-100 recall metric as the SPO tuple scores would be similar (as the object scores are similar). As recall does not check for triplet accuracy, hedging your bets by including multiple triplets with similar scores is a viable strategy. Therefore, in such scenarios choosing top $k$ predictions is often better.
> >
> > 2. Although the authors have given quantitative results on how $C$ improves performance, it is still hard to understand the visual impact of the module. As mentioned in weaknesses point 4, it would be nice to visually see how much of a refinement $C$ provides over the initially estimated bounding box by $H$.
> >
> > 3. There are still certain ablations and experiments missing. See point 6 in weaknesses; point 4 in questions.

---

### Official Review · Reviewer_P8oQ · 2022-07-12

**Rating:** 5
**Confidence:** 4
**Soundness:** 3 good
**Presentation:** 2 fair
**Contribution:** 3 good

**Summary:**

The paper presents a single-stage visual relationship detection method based on a new design of CNN-Transformer network architecture. The main idea is to first compute a shared representation based on the CNN-Transformer encoder, and then to decouple the visual relationship detection into two sequential subtasks: 1) Prediction of the entity-pair location and the predicate class based on a predicate decoder;  and 2) Refinement of the entity locations and prediction their classes given the predicate prediction based on a conditional entity refinement decoder.  A multi-task loss based on Hungarian matching is designed to train two decoders separately, each focusing on its corresponding subtask.  The proposed method is evaluated on the VG150 benchmark with comparisons to the prior methods.

**Questions:**

1. How does the method cope with the potential inconsistency in the SSG prediction?
2. What is the contribution of PnP DETR to the performance improvement?
3. What is the time complexity of the proposed method in comparison to the other baselines?


**Ethics Review Area:**

["I don’t know"]

**Limitations:**

The discussion on the limitations is included in the appendix.

**Strengths And Weaknesses:**

Strengths:
+ The paper presents an efficient single-stage visual relationship detection strategy based on a novel decomposition of the triplet detection task.  The resulting two sequential subtasks have a smaller search space, which simplifies the overall detection task.
+ The proposed method achieves strong results on the VG dataset, outperforming the prior SOTA with more compact network architecture.

Weaknesses:
1) Potential inconsistency in scene graph generation. While the proposed method can produce a set of visual relationship predictions, they are generated conditioned on the predicate estimation. In particular, the locations of entities in each visual relation are refined independently. As a result, two neighboring visual relations (sharing one entity), or two visual relations with the same sub/obj entities can produce two different localizations for the shared entity.
2) Some of the motivation lacks clarity.
+ What is the feature $f_i$ in Line 44? How do "the entity-scene interactions characteristic of each predicate" instantiate in the models described in this paper?
+ Why the matching in the DDTR is computationally expensive? (Line 143)
+ Why the task $Pr(\mathcal{P}|\mathcal{I})$ is a simpler subtask? (Line 241) The variation of the predicates seems to be much larger than entities and they also have more severe long-tail problems.
3) The experimental evaluation is a bit lacking in the following aspects:
+ The paper adopts the PnP DETR as the transformer architecture. It is unclear if this backbone introduces additional benefits in performance, such as entity detection quality.
+ The paper lacks a comparison in time complexity for model inference.
+ In the ablation on the entity refinement module, why use the outputs from the predicate decoder? The details of the evaluation on the task PredDet seem to be missing.

---

> ### Author Response · Authors · 2022-08-02
> **Thanks for your comment**
>
> **Please read the General comments for all reviewers.**
>
> Thanks for providing thoughtful comments. Minor issues will be corrected as suggested. All results and discussions will be added to the final version.
>
> **Potential inconsistency in scene graph generation**
>
> This potential inconsistency is unavoidable for single-stage models (e.g. RelTR, FCSGG) due to the removal of the time-consuming non-maximum suppression. However, this inconsistency should be minimal; otherwise, it will still be penalized by the recall metrics when the bbox IoU is below a threshold (e.g. 0.5).
>
> **What is the feature $f_i$ in Line 44?**
>
> Please refer to the General comments -- Explanations of $f_i$.
>
> **Why the matching in the DDTR is computationally expensive?**
>
> Please refer to the General comments -- Details of baselines.
>
> **Why is the task Pr(P|I) a simpler subtask?**
>
> We did not claim $Pr(\mathcal{P}|\mathcal{I})$ as a simpler task than $Pr(\mathcal{Y}|\mathcal{I})$.
> In L234-L258 of the main paper, we discuss the benefits of the proposed formulation from the probability perspective. Compared to $Pr(\mathcal{P}|\mathcal{Y}, \mathcal{I})$, $Pr(\mathcal{Y}|\mathcal{P}, \mathcal{I})$ avoids the combinatorial space of $\mathcal{O}(\mathcal{E}\times \mathcal{E})$.
>
> **Benefits of PnP-DETR**
>
> Please refer to the General comments -- contributions of PnP-DETR.
>
> **Why use the outputs from the predicate decoder for entity refinement ablation?**
>
> In this ablation, we want to study how much bbox refinement was happening from $\mathcal{H}$ to $\mathcal{C}$. To disregard the result of entity label prediction, we evaluate the model with the PredDet task. For more details, please refer to the General comments -- Effectiveness of the refinement module.

---

### Official Review · Reviewer_2zbH · 2022-07-14

**Rating:** 5
**Confidence:** 4
**Soundness:** 3 good
**Presentation:** 4 excellent
**Contribution:** 3 good

**Summary:**

The paper analyses design choices of one-stage architectures for the task of Scene Graph Generation (SGG) - specifically, the degrees of feature entanglement for predicting predicates (relations) and entities (objects). An observation is made in Fig 1 and Tab 1, that complete coupling of features for predicate and entity classification leads to lower performance, while on the other hand complete decoupling gives better performance but it’s undesirable since it leads to overhead in the form of disjoint queries. The authors propose a mechanism to condition queries for predicting entities with the predicates, which leads to a desired trade-off between completely decoupled and coupled approaches. The authors show improved performance over its counterparts.

**Questions:**

See "Weaknesses" above

**Limitations:**

Yes

**Strengths And Weaknesses:**

## Strengths
- The approach is well-motivated and is communicated clearly

## Weaknesses

- Table 1
    - I couldn’t find the result of TracCQ  here. Is it the same as the last row of Table 2?
    - If yes
        - then is it fair to say that DDTR already outperforms the best performing single-stage approach in Table 2?
        - Also what changes resulted in better performance of DDTR baseline compared to the prior single-stage approaches?
    - Line 294-295: “… It also exhibits a much less drastic drop in the mean recall values when we go from 100 to 20 for mR@K …”
        - In fact, mR@20 scores in Table 1 for SD, DD and DDTR are either comparable or better than the best-performing single-stage approaches. Do the authors have an opinion about this? Are there any specific biases in the paper’s approach that accounts for this behavior?
- Table 3: Correction v/s no-correction
    - The authors mentioned in line 183 that the predicate decoder ($\mathcal{H}$) “… is trained so as to not be penalized for inexact bounding of the subject and object entities. …”
    - But the loss ($\mathcal{L}_p$) in equation 12 includes the term for box loss with no modification for any leeway for box predictions or be penalized lower than what is suggested
    - Moreover, the ablation in Table 3 seems to suggest that when the refinement of boxes is turned off (row 1), the performance doesn’t get affected by a lot and in fact shows better results for mR @ 20. Additionally this also shows up in the visualization of refined v/s original boxes in Supplementary Fig 2, which seems to suggest that no too much correction is happening
    - Does this suggest that the predicate decoder is also predicting boxes as accurately as the final model? Can the authors quantify the correction?
    - Do the authors have an ablation for a variant of the model where the entity refinement decoder only predicts the class of the objects, to really see whether the box refinement is crucial?


### Overall comment
- I like the idea of reducing the search space of entity refinement with the help of predicted predicates, and the fact that preliminary investigation in the form of Table 1 supports the intuition that the authors build on, but the experimental results and ablation could be made stronger and discussed more to substantiate the authors’ claim (given that most of the main paper prioritizes discussing the approach)

---

> ### Author Response · Authors · 2022-08-02
> **Thanks for your comment**
>
> **Please read the General comments for all reviewers.**
>
> Thanks for providing thoughtful comments. Minor issues will be corrected as suggested. All results and discussions will be added to the final version.
>
> **The result of TraCQ compared to baselines in Table 1**
>
> Please refer to the General comments -- Details of baselines.
>
> **Why baselines are either comparable or better than prior single-stage models in mR@20?**
>
> While there are different levels of disentanglement of the task in baselines, there is no sequential dependency between the entity and predicate predictions. However, in prior DETR-based single-stage models (e.g. RelTR, Relationformer), predicate predictions rely on the decoded features of entity queries. This will induce bias during training and potentially become a negative factor when K is low.
>
> **Questions about the entity refinement**
>
> We provide detailed explanations and additional ablation studies to the entity refinement module in the General comments.

---

### Author Response · Authors · 2022-08-02
**General comments for all reviewers (Part 1)**

We thank the reviewers for recognizing that TraCQ is well motivated (2zbH, ZNFu, zNec), clearly presented (2zbH, ZNFu, zNec), novel (P8oQ, ZNFu), and well performed (P8oQ, zNec). The main issues raised by reviewers are clarified as follows.

**Contributions of PnP-DETR**

Due to the limited computing resources, we adopt PnP-DETR [31], which optimizes the computation adaptively with regard to spatial information, resulting in much lower flops (-45\%) and faster convergence. Compared with standard DETR for object detection on the COCO dataset, PnP-DETR achieves a slightly lower AP (42.0 vs 41.8).

**Details of baselines**

Both single decoder (SD) and dual decoder (DD) use a set of shared random queries for entity and predicate predictions (Sec. 3.2 \& Fig. 1), while DDTR decouples these 2 tasks with separate sets of random queries, which requires further matching between the output from 2 sets. Assume $M$ predicates and $N$ entity predictions. There are $M\times \mathbf{P}^N_2$ SPO tuples. To find the best match between 2 sets, a cost matrix is computed for each tuple, which is inefficient (P8oQ) (See inference time in Appendix Table 1). To address this, TraCQ introduces conditional queries to maintain the one-to-one correspondence between the output from 2 sets, which reduces the inference time from 0.22s to 0.15s.

All the baselines and TraCQ generate the same numbers of predicate predictions ($N_p=200$).
However, TraCQ provides the flexibility to refine the entity predictions by implicitly sampling around the bbox predictions from predicate decoder $\mathcal{H}$ with different factors. Specifically, TraCQ generates $N_{ce}$ entity refinements for each bbox prediction in $N_p$ predicate predictions from $\mathcal{H}$, but only chooses the top $k$ SPO-tuple candidates for each predicate (see Sec. 4), which results in $k\times N_p$ predictions. For fair comparison, we adopt the same top $k$ selection for DDTR with $k=5$ (See Table 1 in the main paper). Compared to SD and DD, DDTR and TraCQ have more SPO-tuple predictions when $k=5$. However, TraCQ outperforms SD and DD when $k=1$ by a large margin as well (Table 2 in Appendix).

**Number of parameters**

As shown in Table 2 in the main paper, more parameters do not necessarily lead to better performance. Hence, the performance gain of TraCQ is not the direct outcome of a larger model. In fact, TraCQ is smaller than DDTR (51.2M vs 82.9M) but achieves better performance (mR@20 12.0 vs 9.2). Since it is challenging to design a model of exact same size as the baselines, we ensure each sub-module of the same type (Enc, Dec, FFN) shares the same number of parameters and hyper-parameters across baselines. The number of parameters for SD, DD, DDTR, and TraCQ is 41.7M, 51.1M, 82.9M, 51.2M respectively.

**Number of predictions**

Generating more predictions does not guarantee higher Recall. Regardless of one-stage or two-stage models, the R@K of SGG models is evaluated by ranking all the generated SPO tuples for each image, but only the top K predictions are considered. If the model generates duplicated predictions for a single relation tuple with high detection scores, they will "push" the predictions of other relation tuples out of the top K predictions, which will induce a lower R@K. The model can only benefit from more predictions if the predictions are high-quality and the performance is not saturated. As shown in Table 2 of Appendix, the performance of TraCQ increases from $k=1$ to $k=5$, which shows that the entity refinement module generates high-quality refinements that correct the error of initial predictions. When $k>5$, the performance saturates quickly and there is no benefit to generating more SPO-tuple candidates.

In fact, most prior works generate more predictions than TraCQ. For the models that predict predicates based on entity detection, there are $\mathbf{P}^N_2$ predictions, where $N$ is the number of bbox candidates. The Relationformer [25] mentioned by the reviewer ZNFu is such a case, which has the number of predictions in $\mathcal{O}(N^2)$. However, DDTR and TraCQ only generate number of predictions in $\mathcal{O}(k\times N)$, where $k << N$.

---

> ### Author Response · Authors · 2022-08-02
> **General comments for all reviewers (Part 2)**
>
> **Effectiveness of the refinement module**
>
> The entity refinement module $\mathcal{C}$ not only refines the bbox predictions but also predicts the entity labels. Table 3 in the main paper presents the result of PredDet, where only the ground-truth entity labels are given and the task is to predict $< b_{sub} - p - b_{obj} >$ tuples (L307).
> The input can be thought of as a set of object tags without specifying where they are, and the outputs are the visual relation tuples grounded on the image.
>
> The entity refinement module $\mathcal{C}$ implicitly performs dense sampling around the bbox predictions from $\mathcal{H}$, which may not have a visible impact on bbox refinement when the initial bbox prediction is precise, so the gain is relatively small for PredDet. However, $\mathcal{C}$ is important for learning the entity label distributions and prevents the entanglement of entity feature and predicate feature (See Sec. 3.2). If we remove $\mathcal{C}$ and add a classification head to the predicate decoder $\mathcal{H}$, the model reduces to SD, which underperforms TraCQ. We also conduct a variant suggested by the reviewer 2zbH, where $\mathcal{C}$ only predicts entity labels. The mR@20/50/100 of the suggested baseline and TraCQ are 10.7/11.7/12.1 and 12.0/13.8/14.6, respectively. This indicates that entity bboxes and labels are closely related, and optimizing both training objectives helps the learning of the entity refinement module.
>
> **Explanations of $f_i$**
>
> Our hypothesis is that SGG relies on the entity features $f_e$, the predicate features $f_p$ and the interaction features $f_i$. The entity features locate and distinguish the entities in a scene (ObjDet). The predicate features capture the relationship between the entities (PredCls). The interaction features $f_i$ capture the relation between the entities and their surroundings. The introduction of $f_i$ is motivated by the visualization of the weights from an entity detection model, which shows that the surrounding of an entity is not learned explicitly. $f_i$ is proposed to better estimate the distribution of the background.

---

### Meta-Review · Area_Chair_hwmd · 2022-08-30

**Recommendation:** Accept
**Confidence:** Less certain

**Metareview:**

Paper was reviewed by five reviewers and received: 3 x Borderline Accept, 1 x Borderline Reject and 1 x Weak Accept. Generally, the reviewers thought that the paper was interesting and had merits. Raised issues revolved around (1) lack of clarity in certain parts of exposition; (2) evaluations and ablations that could have been made stronger, and (3) the role of PnP DETR, which isn't a contribution, towards improved performance. Additional reservations dealt with (4)  claims that learning in the non-combinatorial predicate space is easier than the entity pair space, and (5) fairness of comparisons with respect to model capacity and other factors. Authors have provided a compelling rebuttal and this has alleviated many of the reviewer concerns at lest to an extent. Post rebuttal, [ZNFu] remains concerned that generating more predictions may be what is causing improved performance and points out that certain ablations are still missing and can improve the paper. At the same time, [ZNFu], while remains at Borderline Reject, acknowledges that rebuttal has resolved some of the issues in the original review.

AC has carefully considered the reviews, the rebuttal, and the paper itself. This appears to be a rather borderline case, however, considering that the overall sentiment of reviewers is positive and that rebuttal has convincingly addressed an important fraction of concerns raised by [ZNFu] and others, even if not all, it is AC's decision that acceptance of the paper is warranted. Authors are very strongly encouraged to add the ablations, as well as make corrections, suggest by reviewers, for the camera ready.

**Award:**

No

---

### Decision · Program_Chairs · 2022-09-14

Accept